# A Scoping Review of Interventions for Family Bereavement Care during the COVID-19 Pandemic

**DOI:** 10.3390/bs12050155

**Published:** 2022-05-19

**Authors:** Carlos Laranjeira, Débora Moura, Maria Aparecida Salci, Lígia Carreira, Eduardo Covre, André Jaques, Roberto Nakamura Cuman, Sonia Marcon, Ana Querido

**Affiliations:** 1School of Health Sciences of Polytechnic of Leiria, Campus 2, Morro do Lena, Alto do Vieiro, Apartado 4137, 2411-901 Leiria, Portugal; ana.querido@ipleiria.pt; 2Centre for Innovative Care and Health Technology (ciTechCare), Rua de Santo André—66–68, Campus 5, Polytechnic of Leiria, 2410-541 Leiria, Portugal; 3Research in Education and Community Intervention (RECI I&D), Piaget Institute, 3515-776 Viseu, Portugal; 4Nursing Department, Universidade Estadual de Maringá, Av. Colombo, 5790-Zona 7, Maringá 87020-900, Paraná, Brazil; dromoura2@uem.br (D.M.); masalci@uem.br (M.A.S.); ligiacarreira.uem@gmail.com (L.C.); eduardocovre@hotmail.com (E.C.); aejaques@uem.br (A.J.); soniasilva.marcon@gmail.com (S.M.); 5Pharmacology and Therapeutic Department, Universidade Estadual de Maringá, Av. Colombo, 5790-Zona 7, Maringá 87020-900, Paraná, Brazil; rkncuman@uem.br; 6Center for Health Technology and Services Research (CINTESIS), NursID, University of Porto, 4200-450 Porto, Portugal

**Keywords:** bereavement, bereaved family, intervention, scoping review, COVID-19 death

## Abstract

The death of a loved one is a major stressor, and bereaved people are at a higher risk of negative health effects. This risk is higher during the COVID-19 pandemic, which raises the need for understanding existing bereavement support interventions. This scoping review aimed to map and summarize findings from the existing literature regarding bereavement support interventions (i.e., psychosocial and psychotherapeutic interventions) for family carers of people who died of COVID-19. The Arksey and O’Malley methodological framework was used. Five databases—Medline, PubMed, CINAHL, Scopus, and Web of Science—were searched for articles available from the inception of COVID-19 pandemic (March 2020) to January 2022, following the PRISMA guidelines. Among the 990 studies identified, only seven met this study’s inclusion criteria. The analysis comprised three key topics: types of support programmes and bereavement interventions; tools used to measure the outcomes; and evidence of the impacts of the interventions. All studies analysed included interdisciplinary interventions, commonly developed in clinical settings. Support for recently bereaved individuals can entail cognitive behavioural therapy strategies and other tools to educate, guide, support, and promote healthy integration of loss. To mitigate the effects of non-normative family bereavement, we recommend a systematic approach and coordination between organizational settings, including access to informal and professional support, in order to find hope while navigating the aftermath of COVID-19.

## 1. Introduction

COVID-19 is one of the deadliest pandemics in history, with over 468 million confirmed cases and over 6 million fatalities worldwide, as of 22 March 2022 [1]. Behind these numbers are thousands of families who suffer the pain of the loss of loved ones. Death of a loved one, in itself, is usually a difficult process, often permeated by feelings of sadness and anguish. However, in the context of the COVID-19 pandemic, the psychological impacts can be even more intense, as the usual farewell rituals are constrained [2,3,4]. The absence of farewell rituals for COVID-19 fatalities, given the disease’s high transmissibility, has had negative repercussions on the lives of family members, expressed through psychological impairment [5].

Grief generates the need to break affective ties, transform life’s new demands, and establish new relationships, to increase an individual’s resilience [3,6]. Grief is a multidimensional normative process resulting from loss, which involves the adaptation of different aspects and areas of a human being, such as: affective, cognitive, behavioural, and spiritual [7,8]. Worden’s [9] grief model includes the basic tasks that the survivor needs to undertake to adapt to loss, namely: “(1) accept the reality of loss; (2) experience the pain of loss; (3) adapt to a new life without the lost person; and (4) reinvest in the new reality”. They occur in no particular order, although there is a natural order, as completing some tasks presupposes completing other tasks. However, certain tasks may be revisited over time, as grief is a non-linear process (experienced in “waves”, rather than a slow incremental process), and grief tasks have no set schedule [9]. In this regard, Khosravi [10] notes that, during the early months of bereavement, Worden’s task-based model can help healthcare providers support people bereaved due to COVID-19.

Individual and group psychosocial and psychotherapeutic grief interventions have been widely reported for different populations, including violent loss due to homicide or war and more general grief reactions, such as persistent grief distress following the loss of a long-term spouse, grief associated with profession (e.g., first responders or hospice workers), and unexpected loss (e.g., missing person or perinatal loss) [11].

Harrop et al. [12] found that 51% of participants facing high vulnerability to grief exhibited an elevated need for emotional support, particularly in dealing with/expressing emotions and feelings. Most bereaved individuals adapt to loss, but a significant minority report high levels of persistent grief symptoms long after loss [13,14]. Complicated grief is characterized by “excessive rumination, alienation, hopelessness, and intrusive thoughts about the dead” [15] (p. 5). While some individuals experience grief-related depression and post-traumatic stress symptoms, these well-established diagnoses do not adequately capture other complications [13,14]. Several COVID-related variables, including “social alienation, lack of support, and failure to prepare for death”, can contribute to complicated grief [15,16] (p. 5).

Given this scenario, people who lost family members through COVID-19 have an evident need for special care. A recent meta-analysis found that “psychological interventions have a positive effect on pathological grief symptoms” [17] (p. 2). Recent reviews regarding the effect of the COVID-19 pandemic on loss and grief experience found scarce evidence related to bereavement support [12,18]. The current health crisis has raised awareness about the family’s role in people’s lives and the need for health professionals to rethink the family as a focus of care and intervention [19]. Effectively supporting bereaved families requires a grasp of existing bereavement care interventions [20].

A preliminary search was carried out, and no current or ongoing reviews on the subject were found. Therefore, this scoping review aimed to identify published studies that used psychosocial and psychotherapeutic interventions intended to assist family carers to adjust to loss and grief due to COVID-19. We systematically collected and mapped the available literature regarding what was offered to family members bereaving the loss of a relative who died from COVID-19. This review can help healthcare professionals, organizations, and policymakers in their work with bereaved families.

## 2. Materials and Methods

### 2.1. Study Design and Research Questions

We conducted a scoping review, applying the Arksey and O’Malley [21] methodology, which contains five steps: (1) identify the research question, (2) identify the relevant studies, (3) study selection, (4) data mapping, and (5) comparison, summary, and reporting of results. Scoping reviews are useful for mapping key concepts, examining emerging knowledge, identifying knowledge gaps, and reporting available knowledge [22].

The main research questions that guided this scoping review were as follows: “What evidence is there about bereavement support interventions for family caregivers of people who died of COVID-19?” and “What can we learn from the COVID-19 pandemic about the subsequent impact of this type of death on grief?”.

The sub-questions analysed were:-What types of interventions to reduce grief and complicated grief during the COVID-19 pandemic were addressed in the existing literature?-In what settings were these interventions provided?-What were the assessment tools used to assess grief/family grief?-Were there specific components of intervention design (frequency, single or grouped interventions, individual or interdisciplinary application, dose, duration, who delivered intervention, and the application of the intervention) that subsequently influenced outcomes during bereavement?-Were these interventions effective?

This review was prepared based on the Preferred Reporting Items for Systematic Reviews [23] and Meta-Analyses Extension for Scoping Reviews (PRISMA-ScR) [24]. The protocol was registered in the Open Science Framework (OSF—registration number: https://osf.io/bw7fn/ accessed on: 30 March 2022) and was published [25].

### 2.2. Eligibility Criteria

The eligibility criteria were (a) research within the field of health (nursing, psychology, and psychiatry); (b) use of psychosocial or psychotherapeutic interventions, including with bereaved family caregivers (≥18 years) of people who died of COVID-19; (c) in English or Portuguese; and (d) published from March 2020 to January 2022 (this period was chosen to guarantee studies were carried out during the current pandemic). As COVID-19 deaths occur predominantly among older adults, the review also focused “on support for adults or families grieving adult deaths, rather than support for children or families grieving the loss of children” [12] (p. 1167).

All study designs were eligible: experimental and quasi-experimental; observational; qualitative; mixed-method; systematic reviews; meta-analyses; scope reviews; overview articles; and narrative reviews.

### 2.3. Literature Search Strategies

A systematic search was performed in the following databases: Medline, PubMed, CINAHL, Scopus, and Web of Science. Unpublished studies/grey literature were searched in: Open Gray, researchgate.net (accessed on 4 February 2022), europepmc.org (accessed on 4 February 2022), clinictrials.gov (accessed on 4 February 2022), trialregister.net (accessed on 4 February 2022), and Google Scholar (accessed on date month year). Therefore, our approach included a wide range of databases relevant to social and health sciences, and included the grey literature. Medline was searched with a combination of Medical Subject Headings (MeSH) and text words, listed in Table 1, to identify potentially relevant studies. The other databases were searched with the same string, using subject headings and keywords. We conducted database searches on 4th February 2022.

To identify potentially relevant research, two reviewers (D.M. and E.C.) separately examined the titles and abstracts of all publications. Following a study of the full text, two of the three reviewers (C.L., A.Q., and D.M.) independently assessed eligibility. Disagreements were settled by discussion and, if required, arbitration by a third party.

### 2.4. Data Charting and Summarizing Data

A data extraction form derived from the Joanna Briggs Institute [26] was used to map the following information from each article included in the review: author(s), year and country of study, intervention, study design, study goals, sample characteristics, outcomes/methods, and any other key findings related to the research questions. One reviewer (D.M.) extracted data, while a second reviewer (E.C.) checked the initial extraction against the original article, adding data when necessary. Data from papers reporting the same study were combined in a single data extraction form. Discrepancies in data were resolved through discussion, with arbitration by an independent reviewer, if necessary.

The quality of the studies was not evaluated during the scoping review because the goal of a scoping review is to identify gaps in the literature and propose potential research questions for systematic review [21].

### 2.5. Ethical Considerations

Ethical approval and patient consent were not mandatory.

## 3. Results

### 3.1. Study Selection and Characteristics

The initial search identified 990 potential articles. After six duplicates were removed, 984 articles were reviewed for the appropriateness of their titles and abstracts, which resulted in the removal of 950 articles. Thirty-four full-text articles were assessed by D.M., in consultation with the second author (E.C.), for eligibility using the inclusion and exclusion criteria; 27 articles that did not meet the inclusion criteria were excluded from the scoping review. After a three-part screening process, seven studies were included. The flow diagram is presented in Figure 1.

We included one descriptive cross-sectional study [27]; four RCT studies [28,29,30,31], three which were in the protocol phase; one qualitative study [32]; and one letter to the editor with action research data [33]. Two studies were conducted in Italy. The other five were conducted or planned in the Netherlands, France, Germany, Mexico, and China.

### 3.2. Research Key Topics

Summaries of the results of the included articles are presented in Table 2. These studies differed in the target population and type of bereavement, where and who delivered the intervention, the methodological approach, and the intended outcomes.

#### 3.2.1. Types of Support Programs and Bereavement Interventions

The studies included in this review reported on the support programs and bereavement interventions launched in response to the COVID-19 pandemic. The interventions focused on psycho-educational strategies, risk assessment of prolonged grief disorder (PGD) and other mental health problems (e.g., PCBD and PTSD), and provided specialized care to individuals with complex needs. All interventions exclusively targeted bereaved adults, with sample sizes ranging from 21 to 246 subjects.

Most interventions and programs involved local support, commonly in clinical settings. Across all programs, support was delivered by psychologists, psychiatrists, therapists, and nurses. Staff commitment and a structured plan were key positive elements for the bereaved. The intervention programs presented individual-based counselling and support sessions by phone or online. Assistance was offered individually because the newly bereaved are usually not prepared for group sharing until several months have passed [12]. Some studies included intervention by phone [27,32], information provision [29,30], grief counselling [27,28,31], psychological support, and specialized mental healthcare for high-risk situations [32,33]. Each strategy was implemented singularly or in association with other strategies.

Three studies provided cognitive behavioural therapy (CBT) interventions, whose goals centred around change, not around “recovery” (with the intention of returning to the pre-bereavement stage). The online multi-component psychological intervention [27] involved CBT sessions focused on mindfulness techniques, behavioural activation therapy, and positive psychology. The online grief counselling plan involved 8–10 sessions focused on “(1) understanding and accepting grief reactions; (2) managing painful emotions; (3) learning to care for yourself; (4) increasing contact with others; (5) coping with difficult days; and (6) adapting to a new life” [31] (p. 8). It also included information on conventional grieving processes and traumatic grief symptoms, as well as strategies for dealing with loss and resuming previous life activities. The grief-specific online CBT [29] and the online unguided grief-specific CBT intervention [30] proposed psychoeducation, cognitive restructuring assignments, exposure, and behavioural activation assignments.

Overall, the interventions started in the immediate aftermath of loss, and lasted for between three and six months after the event.

#### 3.2.2. Outcome Measurement Tools

To document the impact of bereavement interventions, several assessment measures were proposed:(a)Bereavement risk assessment following a patient’s death—Inventory of Traumatic Grief (ITG) and the Grief-related Avoidance Questionnaire (GRAQ) to evaluate grief symptoms and avoidance behaviours in the bereaved [31];(b)Complicated or prolonged bereavement assessment following a patient’s death—Prolonged Grief Questionnaire (PG-13) [31], an algorithm for diagnostic criteria for PGD; Traumatic Grief Inventory—Clinician Administered (TGI-CA) to assess Persistent Complex Bereavement Disorder (PCBD) symptoms [29,30]; PTSD Checklist for DSM-5 (PCL-5) and Post-Traumatic Stress Disorder Symptom Scale to evaluate PTSD symptoms [28,29,30,31]; and the Inventory of Complicated Grief [27] to assess indicators of pathological grief, such as anger, disbelief, and hallucinations;(c)Psychiatric symptoms assessment in bereaved families (e.g., anxiety, stress, and depression)—Patient Health Questionnaire (PHQ-9) [29]; Center for Epidemiologic Studies Depression Scale (CES-D) [28]; Depression Anxiety and Stress Scale (DASS-21) [28,31]; the Generalized Anxiety Disorder 7-Item (GAD-7) scale [28]; Typical Beliefs Questionnaire (TBQ) [31] to measure maladaptive cognitions common in the bereaved; and the Plutchik Suicide Risk Scale and the Scale for Suicidal Intention (SSI) to measure the severity of suicidal intentions [28,31].

#### 3.2.3. Evidence on the Impacts of Interventions

Studies varied in psychotherapeutic approaches and therapists, and rarely reported quantifiable outcomes, hindering efforts to compare the effectiveness of the interventions. A single RCT study [30] reported the effect of a bereavement intervention on two core outcomes: Persistent Complex Bereavement Disorder (PCBD) and Post-Traumatic Stress Disorder (PTSD). In the evaluation of the unguided online behavioural therapy [30], bereaved individuals receiving the online CBT treatment showed significantly lower PTSD symptom levels in post-treatment, compared to the control group. The analysis found a significant effect of the online grief-specific CBT-intervention on PCBD and PTSD symptom levels, with a stronger effect for PCBD. The results provide valuable knowledge regarding the effectiveness of online brief-specific CBT interventions because there is no previous evidence-based research.

Three of the studies under review are underway and results are not available currently. In the other studies under analysis, the participant feedback was mostly collected using qualitative data. This feedback provides insights into their lived experiences, feelings of connectedness, and expectations for the future [27,32,33].

In general, there is scant evidence from the included studies referring specifically to the effectiveness of bereavement interventions, because most of them are in the protocol phase. However, several studies refer that supporting protective factors (e.g., resilience traits, coping strategies, life satisfaction, and personal resources) for the bereaved’s overall mental health and socioemotional wellbeing are encouraging intervention approaches [27,28,33].

## 4. Discussion

The COVID-19 pandemic has had an unprecedented effect on economic, social, and healthcare systems worldwide, but the impact on mental health status and bereavement care has not yet been accessible [12]. However, there is consistent evidence that COVID-19 sudden deaths entail high-risk factors for poor bereavement outcomes, namely PGD, PTSD, and low mental health status [7,12,34,35,36].

As mentioned above, there is scarce evidence about the effectiveness of any approach/intervention to decrease stress and suffering after the death of a loved one, shorten the length of the normal process, or prevent long-term negative consequences [37,38]. None of the analysed studies provide high-quality evaluation of the program to enhance health outcomes. Although there is some evidence of personal loss and grief responses to the pandemic, there is little evaluation “of what constitutes an effective systems approach to bereavement service delivery in this context” [12] (p. 1177). Interventions should focus on an integrated care model to capture the components and processes involved. The design and implementation of such interventions need to address people who, in spite of suffering, might feel reluctant in asking for help.

Several key services were identified across interventions: proactive outreach; organization of local support; training in crisis management competencies; psychoeducational intervention, as well as group-based support modalities; risk assessment of prolonged grief disorder; and specialized mental health referral [12,39].

Cognitive-behavioural techniques can assist the bereaved in managing barriers to natural grieving by re-experiencing loss and sentiments toward the departed and therefore reshaping the meaning of loss [39]. The education and training of healthcare professionals, as well as a needs assessment of individuals, usually determines the type of assistance that is suggested. According to Harrop et al. [12], although professionals vary in their individual methods and approach, they should share common characteristics, namely: non-judgmental compassionate care and a willingness to assist the bereaved in restoring normal function and perception of socioemotional well-being.

Given that we are presently confronted with a worldwide sanitary crisis, it is possible that several predictors of complicated grief are potentiated by the COVID-19 pandemics, such “as lack of readiness for death, high stress at the time of death, preventable death, and poor perceived social support after the death” [40] (p. 16). Thus, more attention should be given to this issue, because a huge number of survivors are expected to adopt this style of grieving in the coming years [40].

Following a loss, everyone grieves and needs care, reassurance, and customized information. The imposed COVID-19 control measures determined alternate strategies of remote delivery, such as phone/virtual therapy, virtual reality groups, online discussion forums, or even outdoor activities [12]. Furthermore, providing CBT online might reduce treatment expenses, making treatment more accessible to people in need [17,38]. However, one of the major issues with re-evaluating online interventions is their broad presentation of theoretical content and weak explanation of the key design elements of the human–computer interaction [41]. There is an urgent need for evidence on their feasibility, effectiveness, and acceptability due to a dearth of research on these strategies in catastrophe and conventional bereavement situations [12].

### 4.1. Implications and Suggestions for Research and Practice

This scoping review’s findings underline the benefits of evidence-informed bereavement care to better support families, focusing mainly on the individual needs of the bereaved person. Furthermore, strategies that are family-centred and entail peer support interventions and group work have been demonstrated to be beneficial [42]. In addition, healthcare institutions play an important role as educators on mental health and wellbeing in a broad sense, and as facilitators of support for those bereaved people demanding more differentiated support.

Bereaved individuals were approached, mainly, by web-based interventions. This excludes individuals who do not have access to the internet, and certain groups (e.g., the elderly) were not approached by researchers or were reluctant, as the intervention was provided online. This questions whether the results are generalizable.

Additionally, there is a definite need for robust primary research on grief experiences and bereavement support due to COVID-19. Overall, there is a scarcity of data on effective intervention models. Nonetheless, the research reviewed here emphasizes the need for primary intervention approaches. An example of good practice is the incorporation of grief training into health education. For that purpose, health schools need to provide bereavement training, such as death education, socioemotional regulation practices, and counselling in grieving.

This review also offers suggestions that professionals and healthcare providers can adopt before and after a patient’s death to better mitigate a complicated grief reaction for bereaved families in the COVID-19 times. According to Carr et al. [43], “small-scale interventions may be effective in mitigating bereavement symptoms, at least in the immediate aftermath of loss” (p. 428). Nevertheless, “major investments in social programs and infrastructures are required in the longer term” [43] (p. 428). Further investment in providing “tailored bereavement support is needed to meet the diverse needs and backgrounds of bereaved people, including support that is culturally and crisis/context competent, and group-based support for those with shared experiences and characteristics” [12] (p. 1986). It is also essential to recognize the mediators affecting the grief tasks outlined by Worden, to understand how people cope with COVID-19-related bereavement [10].

Informal community-based programs can help counteract isolation, while longer-term educational and societal initiatives can foster community support for the bereaved [11]. Post-bereavement mutual support groups are an excellent example of how individuals may cope with personal grief, bereavement-related challenges, and the reorganization of their lives. In addition, we recommend the development of an international data source that includes evaluation approaches and methods used in the field of bereavement care for evidence-informed practice.

With these tips in mind, several areas should be studied further and implemented in different settings (e.g., mental health centres, primary care, and/or hospitals):Comparing internet-based versus in-person support for the bereaved aimed at different stages of the grieving trajectory (e.g., immediate or long-term adjustment);Comparing interventions with high- vs. low-risk people, including the risk management process;Assessing the effectiveness of various psychosocial and psychotherapy interventions in reducing psychologic distress and enhancing social functioning;Comparing the benefits of individual vs. group support during the different stages of the grieving process;Establishing a link between self-reported and objectively assessed outcomes;Comparing the effectiveness of different types of interveners;Determining how information regarding bereavement processes influences professional behaviour as well as the behaviour, grieving process, and results of bereaved people.

### 4.2. Study Strengths and Limitations

We adopted a rigorous approach to extract, search, and appraise the existing literature; however, the review approach also had limitations, including: short evaluation follow-up times or cross-sectional studies; a lack of appropriate measures for preventative interventions; variation in outcome measurement tools; low sample numbers, common in empirical studies; difficulty separating findings by age and grief stage; lack of statistical analysis; inclusion of few articles in the review; and inability to answer all of the research questions because of a scarcity of information in the protocols. As a result, it was difficult to directly compare the findings of the included studies and assess the generalizability and transferability of the results. Our option of including studies in the protocol phase was due to the lack of perspective, research, and information about how to deal with the aftermath of COVID-19. The present review aims to contribute to this field. Furthermore, because we excluded non-English and non-Portuguese studies, our results may be biased.

Several studies focused particularly on the bereavement of intimate relatives, and not bereavement as a broader human experience. A restricted emphasis on the death of a loved one may limit our knowledge of successful interventions as well as our wider grasp of grief and loss concepts. In addition, because bereaved families were only tracked for a few months, the long-term impacts of bereavement are unknown. However, this scoping review is a significant step toward mapping the impact of family bereavement support due to the COVID-19 pandemic and indicates what research deserves further study.

## 5. Conclusions

Given the lack of specific evidence, this review’s conclusions in relation to COVID-19 are narrow. Although experiencing bereavement is common, the specific experience of loss and bereavement due to the COVID-19 pandemic may have been potentiated by the inability to be near the dying person, bereaving with family and friends, or attending farewell rituals and funeral ceremonies. This scenario amplifies the psychological distress and mental health suffering of bereaved people. Because the experience of proximity to death has been disturbed, evidence supports several intervention options (single or grouped), ranging from individual-based counselling to support sessions by phone or online and professionally led by psychologists, psychiatrists, therapists, and nurses commonly provided in clinical settings. Notwithstanding, more primary studies of grief experiences during pandemics and how bereavement services and health systems react to meet their needs are required.

## Figures and Tables

**Figure 1 behavsci-12-00155-f001:**
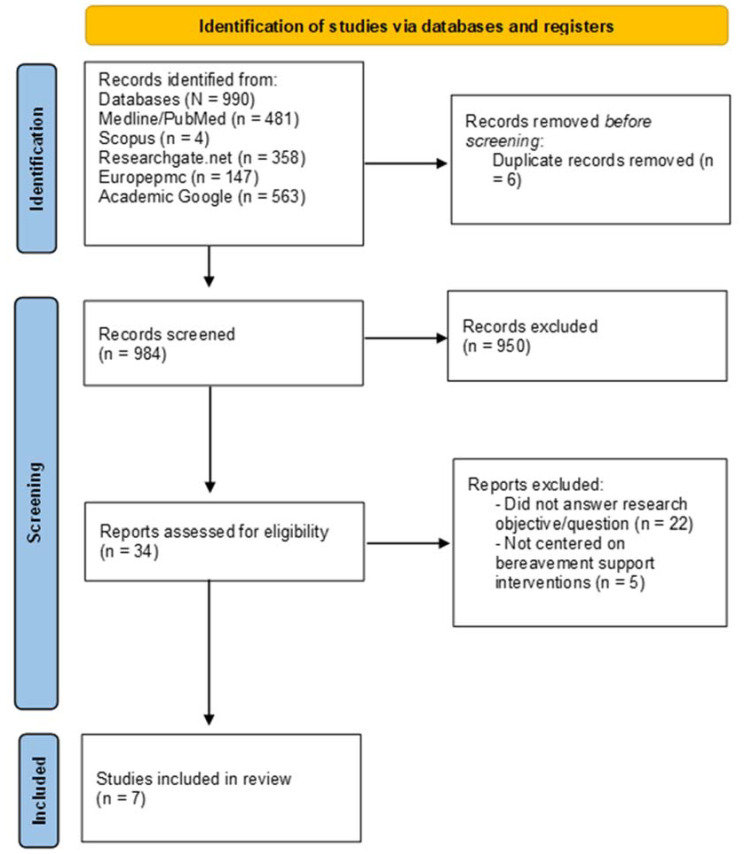
Flow diagram of the results of the literature search.

**Table 1 behavsci-12-00155-t001:** Search strategy used in Medline for identifying potentially pertinent articles.

Key Concepts	TITLE-ABS-KEY
	((“family grief *”) OR (“caregiving grief *”) OR (“bereave *”) OR (“mourn *”) OR (“grieve”) OR (“grieving”) OR (“widow”))
AND	((“psychosocial intervention”) OR (“psychotherapeutic intervention”) OR (psychology *) OR (“psychotherapy *”) OR (“counselling”) OR (“therapeutic alliance”) OR (“social support”) OR (“self-care”) OR (“self-management intervention”) OR (“e-health”) OR (“education *”))
AND	((“pandemic *”) OR (“epidemic *”) OR (“COVID-19”) OR (Corona *) OR (“2019-nCoV”) OR (“SARS-CoV-2”))

* Truncation in keyword searching.

**Table 2 behavsci-12-00155-t002:** Study characteristics and results.

Authors/Country	Intervention	Study Design	Study Objectives	Sample Characteristics (*n*)	Outcomes/Methods	Key Results
Borghi et al. [27]—Italy	Phone-based psychological intervention [27]	Descriptive and cross-sectional study	Describe the experience of a Clinical Psychology Unit in Milan that provided a phone-based early psychological intervention to families of hospitalized COVID-19 patients who died during the pandemic’s first wave [27].	“284 families were called, and 246 family members received the intervention (38 family members were unreachable)” [27] (p. 3).	Phone calls acted as a psychological intervention to help all families, while also assessing psycho-emotional issues and risk factors that needed more specialized care. The “assessment of risk and protective factors” determined the presence or absence of factors affecting grief, sustaining “coping strategies and resources” from family members, and shaping “how much the complex interplay of very early risk and protective factors may have potentially affected the normal bereavement process in each family member” [27] (p. 4).	Written reports were reported after each call. “Most bereaved family members felt grateful for the call and support”. After this assessment, the family member evaluated to be at risk for grieving difficulties was offered the possibility of a referral for “further psychological support” [27] (p. 4).
Mallet et al. [33]—France	The Support Intervention for Bereavement (SIB) involved screening for risk factors in caregivers (CG). Family members could have a one-time intervention or long-term follow-up. The intervention was always individual (one on one). The authors created a hotline number with a dedicated phone (available every weekday, 9am–5pm, with the possibility to record a message). They also created a messaging group to facilitate communication, and produced a letter introducing the SIB, with contact information and condolence letter [33].	Case study with action research approach	Apply a creative telehealth solution that supports families and provides bereavement care [33].	After screening, the nurse called 15 relatives (13 bereaved; 2 relatives of patients who were not dead (but with a high mortality risk)), sometimes more than once. The mean duration of each call was thirty minutes. The SIB was also contacted for possible situations leading to death (with survival at the end). Six of the thirteen bereaved contacts were followed up at least four times, for forty-five minutes calls [33].	First-line intervention (outcomes) —risk factors for complicated grief; —symptoms of acute grief; —adaptive strategies and resilience traits. Second-line intervention Of the 13 bereaved initial contacts, 6 were referred for psychological follow-up with SIB volunteers. This intervention was available according to the situation, discussed by the SIB. With the relative’s authorization, a referral psychologist or psychiatrist could call the bereaved and propose a short follow-up (maximum of 4 calls, mean call duration of 45 min). A cognitive behavioural therapy (CBT) model was used, as it provides a specific framework to treat complex grief reactions [33].	In all situations, the relative requested help; 38.4% of the bereaved contacts also asked for help in the funerary rituals in the context of COVID-19; 31% also sought help with announcing the death to other family members. None of them had the opportunity to say goodbye. The complaints mainly concerned sleep disorders, anxiety about isolation linked to COVID-19, and loss of appetite. They indicated no suicide intent, but half of them reported a heightened sense of guilt. Two grieving people felt stigmatized and embarrassed to reveal the cause of death to a friend [33].
Menichetti Delor et al. [32]—Italy	Phone call intervention—The main goals were: (a) to support the family by providing a safe space for them to express their loss-related emotions; (b) to verify and sustain spontaneous psycho-emotional resources; and, finally, (c) to refer for additional psychological support if the psychologist observed highly complex/at-risk situations [32].	Qualitative approach	Evaluate the contents and functions of this early psychological phone follow-up by the clinical psychologists who participated in the calls [32].	Over the course of three months, 246 families were contacted. Following each conversation, psychologists completed written reports. Such reports described the contents of the conversation and provided further information on the timing and position of the interlocutor within the family [32].	The following topics were investigated: (i) family experiences and needs that surfaced during the calls; (ii) family solutions in place to cope with loss; (iii) activities taken by the psychologist during the call; and (iv) roles played by the calls according to the interviewee [32] (p. 11).	The findings revealed a convergence in initial reactions of loss and trauma, which was worsened by characteristics specific to the present “emergency scenario, such as a lack of protective factors (e.g., social support, life chances) and the presence of shared precipitating/perpetuating causes (e.g., isolation, feelings of guilt, lack of farewell rituals)” [32] (p. 10).
Schrauwen [30]—Netherlands	The intervention consisted of an eight-week unguided online behavioural therapy (including exposure, cognitive restructuring, and behavioural activation) [30].	RCT	Investigate the efficacy of an online grief-specific cognitive behavioural therapy (CBT) intervention for bereaved individuals during the COVID-19 pandemic [30].	Eligible participants were randomly attributed to either the treatment group (N = 21) or waitlist-control group (N = 32) [30].	Persistent Complex Bereavement Disorder (PCBD), post-traumatic stress disorder (PTSD), and depression symptom severity were assessed during (1) pre-treatment/pre-waiting period and (2) post-treatment and/or post-waiting period. Assessment consisted of clinical telephone interviews [30].	Analysis found a significant effect of the online grief-specific CBT-intervention on PCBD and PTSD symptom levels, with a stronger effect for PCBD [30].
Dominguez-Rodriguez et al. [28]—México	Online multi-component psychological intervention (based on cognitive behavioural therapy (CNT), mindfulness, behavioural activation therapy (BAT), and positive psychology (PP)) [28].	RCT—study protocol	Provide a self-administered intervention consisting of 12 sessions based on CBT, mindfulness, BAT, and PP, with the goal of lowering the risk of developing Complicated Grief Disorder (CGD), especially from the COVID-19 contingency, and improving quality of life [28].	Eligible participants: - Adults (>18 years old) who had suffered the loss of a loved one in a period no longer than 6 months prior to the study; - Symptoms of general anxiety disorder and/or depression and/or, grief symptoms; - Access to a communication device with access to the internet [28].	Intended outcomes: Improved life satisfaction and quality of life are expected following the conclusion of the intervention. After the intervention, anxiety and depression symptoms are expected to decrease, and sleep quality to improve. Such changes are likely to last between 3 and 6 months after the intervention procedure is completed [28].	Not reported
Tang et al. [31]—China	Online grief counselling program [31]	RCT—study protocol	Investigate the mental health of bereaved people during the COVID-19 pandemic, train grief counsellors to support the bereaved people, and assess the effectiveness of grief counselling [31].	“160 Chinese bereaved people will be recruited online. Participants in this research must be over the age of 18 and have lost first-degree relatives during COVID-19 [31] (p. 8).	Intended outcomes: “reducing prolonged grief symptoms, post-traumatic symptoms, depression levels, and suicidal intentions in bereaved individuals” [31] (p. 8).	Not reported
Reitsma et al. [29]—Germany	Grief-specific online CBT (based on psychoeducation; cognitive restructuring assignments; exposure; behavioural activation assignments) [29]	RCT—study protocol	Examine the effectiveness of grief-specific online CBT in lowering PCBD, PTSD, and depressive symptoms in bereaved people due to COVID-19 pandemic [29].	Participants are people who lost a loved one at least 3 months earlier during the COVID-19 pandemic with clinically relevant levels of PCBD, PTSD, and/or depression [29].	Intended outcomes: decreasing symptom-levels of PCBD, PTSD, and depression [29].	Not reported

## Data Availability

Not applicable.

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
