# Peer review of "A Scoping Review of Interventions for Family Bereavement Care during the COVID-19 Pandemic"

_behavsci, 2022, doi:10.3390/bs12050155_

Round 1
Reviewer 1 Report
A review of literature is conducted and introduced in the manuscript A scoping review of interventions for family bereavement care 2 during the COVID-19 pandemic. The aim of the review is to map findings regarding bereavement support interventions for family carers of people who died of COVID-19. The manuscript is clearly written and the subject matter is important. However, there are several points that need attention and editing before the publication of the manuscript, for example in terms of the relation between the research questions, reviewed literature and the findings. In general, it would strengthen the manuscript and its argument if the subject was explored in more depth.
- Research questions and findings:
Lines 88-101
Regarding the following research questions which are posed:
There needs to be a clearer explanation and more discussion about the findings in relation to this research question in the context of reviewed articles:
"What can we learn from the COVID-19 pandemic about the subsequent impact of this type of death on grief?"
In the manuscript, the word “of” is missing from the above question.
In relation to this research question, this conclusion is stated on lines 350-355.
“While experiencing bereavement is common, the specific experience of loss and bereavement due to the COVID-19 pandemic may have been potentiated by the inability of being near the dying person, bereaving with family and friends, or attending farewell rituals and funeral ceremonies.”
What was found within the reviewed literature in relation to this conclusion?
“What types of interventions to reduce grief and complicated grief during COVID-19 pandemics were addressed in the existing literature?”
The types of interventions are discussed and mentioned in various places within the manuscript:
Abstract
Line 18: “…bereavement support interventions…”
Lines 25-26: “…Cognitive Behavioural Therapy strategies and other tools to educate, guide, support and promote healthy integration of loss.”
Introduction
Line 75: “…psychosocial and psychotherapeutic interventions…”
Table 2 provides more detailed information about the types of intervention.
In section 3.2.1 there is also detailed information provided about the different types of interventions applied.
An integrated conclusion about the types of interventions is provided in lines 271-274:
“Although professionals vary in their individual methods and approach, they share common characteristics: non-judgmental compassionate care, as well as a willingness to assist the bereaved people in restoring normal functioning and a perception of socioemotional well-being [11].”
It would be helpful to make reference to the way in which this is stated in the reviewed articles in order to explain how you came to this conclusion.
I also suggest that you coordinate and make clearer within the abstract and introduction which types of interventions are being reviewed.
“In what settings were these interventions provided?”
Information about the settings can be found in Table 2. It would be helpful to provide an overview and summary of the findings for this research question in the conclusion.
“Were there specific components of intervention design (frequency, single or grouped interventions, individual or group application, dose, duration, who delivered intervention, application of the intervention) that subsequently influenced outcomes during bereavement?”
“Were these interventions effective?”
This is discussed in section 3.2.3. where the findings of different studies are reviewed. If the former research question remains as it is, those findings need to be discussed in relation to the factors stated: “frequency, single or grouped interventions, individual or group application, dose, duration, who delivered intervention, application of the intervention.”
---
- There are a few places within the manuscript where the findings are generalized and not necessarily related to the findings of the review or the connection is not clear. It is suggested to make those general statements more connected to the research questions, literature review and specific findings in that regard.
Lines 246-247
“However, several refer that supporting protective factors for the bereaved’s overall mental health and socioemotional wellbeing are encouraging intervention approaches.”
Which ones are those approaches and how did the authors come to this conclusion?
Lines 260-262
“Interventions should focus on a public health strategy rather than a clinical model, since at-risk persons are not always recognized in advance and people appear reluctant to ask for help.”
In which reference is this stated and how did the author(s) come to this conclusion? The line of thought and logic of the sentence is also difficult to comprehend. “Interventions should focus on a public health strategy rather than a clinical model...” Does this need to be a choice, can’t they go together? I don´t understand the link here that says that it is more important to focus on a public health strategy rather than a clinical model because “... persons are not always recognized in advance and people appear reluctant to ask for help”.
Line 296
“Bereaved individuals were approached, mainly, by web-based interventions. This excludes individuals who don’t have access to the internet, and certain groups (e.g. the elderly) weren’t approached by researchers or were reluctant, as the intervention was provided online. This questions whether results are generalizable.”
There are also other reasons why the results may not be generalizable, for example lack of statistical analysis and few articles being reviewed. I suggest that if the results are not considering generalizability, there is more of a need to go deeper into the function of each method and how it was received by the user.
----
I question why the three articles which do not yet have findings are included. How do they answer the research questions and contribute to the findings of the manuscript? Why is it important to include those articles? How do they contribute to the findings of the review?
Author Response
Dear Reviewer,
We appreciate your suggestions very much, we tried to respond to all of them including some new content in the paper, as we consider the suggestions very pertinent to improve the quality of our paper. The changes in the revised manuscript are highlighted in grey.
Reviewer 1
> Research questions and findings:
Lines 88-101
Regarding the following research questions which are posed:
There needs to be a clearer explanation and more discussion about the findings in relation to this research question in the context of reviewed articles:
"What can we learn from the COVID-19 pandemic about the subsequent impact of this type of death on grief?"
In the manuscript, the word “of” is missing from the above question.
In relation to this research question, this conclusion is stated on lines 350-355.
“While experiencing bereavement is common, the specific experience of loss and bereavement due to the COVID-19 pandemic may have been potentiated by the inability of being near the dying person, bereaving with family and friends, or attending farewell rituals and funeral ceremonies.”
What was found within the reviewed literature in relation to this conclusion?
“What types of interventions to reduce grief and complicated grief during COVID-19 pandemics were addressed in the existing literature?”
The types of interventions are discussed and mentioned in various places within the manuscript: The types of interventions are discussed and mentioned in various places within the manuscript:
Abstract
Line 18: “…bereavement support interventions…”
Lines 25-26: “…Cognitive Behavioural Therapy strategies and other tools to educate, guide, support and promote healthy integration of loss.”
RESPONSE:
Thank you for pointing this out. We agreed with your comment, for that reason some of the learnings that the pandemic brought about the theme of mourning were clarified in the discussion. Additionally, and in the conclusion, the text was reformulated in order to reflect the main results obtained. As for bereavement interventions, these were grouped in the form of psychosocial and psychotherapeutic interventions in order to respond to the diversity of existing strategies. Semantic and grammatical inaccuracies have been corrected.
2) “Although professionals vary in their individual methods and approach, they share common characteristics: non-judgmental compassionate care, as well as a willingness to assist the bereaved people in restoring normal functioning and a perception of socioemotional well-being [11].”
It would be helpful to make reference to the way in which this is stated in the reviewed articles in order to explain how you came to this conclusion.
I also suggest that you coordinate and make clearer within the abstract and introduction which types of interventions are being reviewed.
“In what settings were these interventions provided?”
Information about the settings can be found in Table 2. It would be helpful to provide an overview and summary of the findings for this research question in the conclusion.
“Were there specific components of intervention design (frequency, single or grouped interventions, individual or group application, dose, duration, who delivered intervention, application of the intervention) that subsequently influenced outcomes during bereavement?”
“Were these interventions effective?”
This is discussed in section 3.2.3. where the findings of different studies are reviewed. If the former research question remains as it is, those findings need to be discussed in relation to the factors stated: “frequency, single or grouped interventions, individual or group application, dose, duration, who delivered intervention, application of the intervention.”
RESPONSE:
Thank you for your insightful suggestions. We agree with the reviewer, so we clarify in the abstract which types of interventions are being reviewed and what settings were these interventions provided. We chose to present a broad discussion on the characteristics of the interventions, given the scarcity of information. Justifying for this reason there is not an individualized answer to each of the questions.
3) There are a few places within the manuscript where the findings are generalized and not necessarily related to the findings of the review or the connection is not clear. It is suggested to make those general statements more connected to the research questions, literature review and specific findings in that regard.
Lines 246-247
“However, several refer that supporting protective factors for the bereaved’s overall mental health and socioemotional wellbeing are encouraging intervention approaches.”
Which ones are those approaches and how did the authors come to this conclusion?
RESPONSE:
Thank you again for pointing this out. We agreed with your suggestion, and additional information has been added to support this conclusion.
4) Lines 260-262
“Interventions should focus on a public health strategy rather than a clinical model, since at-risk persons are not always recognized in advance and people appear reluctant to ask for help.”
In which reference is this stated and how did the author(s) come to this conclusion? The line of thought and logic of the sentence is also difficult to comprehend. “Interventions should focus on a public health strategy rather than a clinical model...” Does this need to be a choice, can’t they go together? I don´t understand the link here that says that it is more important to focus on a public health strategy rather than a clinical model because “... persons are not always recognized in advance and people appear reluctant to ask for help”.
RESPONSE
Ok! We agreed with your comment, so we reorganized the sentence.
5) Line 296
“Bereaved individuals were approached, mainly, by web-based interventions. This excludes individuals who don’t have access to the internet, and certain groups (e.g. the elderly) weren’t approached by researchers or were reluctant, as the intervention was provided online. This questions whether results are generalizable.”
There are also other reasons why the results may not be generalizable, for example lack of statistical analysis and few articles being reviewed. I suggest that if the results are not considering generalizability, there is more of a need to go deeper into the function of each method and how it was received by the user.
----
I question why the three articles which do not yet have findings are included. How do they answer the research questions and contribute to the findings of the manuscript? Why is it important to include those articles? How do they contribute to the findings of the review?
RESPONSE
Thank you for pointing this out. We expanded the limitation subsection with more information that supports the decision of integrating the study protocols. In addition, we added more potential biases to our scoping review.
We hope we have accomplished all the suggestions to consider this paper to be published. Thank you again for helping us to improve our work.
Respectfully,
The Authors
Reviewer 2 Report
- Typo line 75.
- Are we mixing terms in a confusing way? Complicated Grief vs PGD?
- I wonder if your exclusion criteria were too limiting?
- I enjoyed reading this article. It flows well and is well written. I believe that CBT is brought up without enough evidence of its efficacy. Finally, the article leaves one feeling frustrated. You have taken 14 pages to tell us that, with regard to Covid-19 and bereavement, we have no idea what we are doing. I agree with this conclusion, I just find it frustrating nevertheless.
Author Response
Dear Reviewer,
We appreciate your suggestions very much, we tried to respond to all of them including some new content in the paper, as we consider the suggestions very pertinent to improve the quality of our paper. The changes in the revised manuscript are highlighted in grey.
Reviewer 2
> Typo line 75.
Are we mixing terms in a confusing way? Complicated Grief vs PGD?
I wonder if your exclusion criteria were too limiting?
I enjoyed reading this article. It flows well and is well written. I believe that CBT is brought up without enough evidence of its efficacy. Finally, the article leaves one feeling frustrated. You have taken 14 pages to tell us that, with regard to Covid-19 and bereavement, we have no idea what we are doing. I agree with this conclusion, I just find it frustrating nevertheless.
RESPONSE
Thank you for pointing this out. The issues raised by the reviewer have been revised. Added more detail about the concept of complicated grief in the introduction. We explain in the limitations subsection the risk of biases associated with our inclusion/exclusion criteria. We also feel the same when we finished the review. In this sense, is very important to address more primary studies on the theme.
We are deeply grateful for the time and effort you have invested in reviewing our work, and we certainly remain open to any further suggestions for the continued improvement of this paper.
Respectfully,
The Authors
Reviewer 3 Report
This is a paper summarizing existing scientific evidence about the bereavement support interventions for family carers of people who died because of the COVID-19 infection. The review is of good quality, and it is of interest for the readers; however, several minor changes are proposed to improve its quality.
In the abstract section, it is not clear how many studies are reporting individual vs. group interventions for such families affected by death of a family member by COVID-19.
The introduction section is brief. I would recommend to expand it by adding some research about the efficacy of bereavement groups or individual interventions in other contexts. For instance, there is a high amount of evidence of evidence in the perinatal period and other losts.
In the methods section, the authors reported thay they conducted a scoping review according to Arksey and O'Malley and they describe the five steps and the main research questions. PRISMA guidelines have been updated. I would recommend to use guidelines from 2020. The flow chart diagram is also modified according to the 2020 version. It would improve the quality of the paper.
ALthough there is few data on the topic, the present review is really relevant. Future studies should be designed. The authors, at the end of the discussion section reported that "With these tips in mind, several areas should be studied further". How would the authors design these interventions? Should be implemented in mental health centers, in primary care or in hospitals?
There are few data on the topic. I would recommend to propose an international data source to include all the information about efficacy of these interventions. This can be also concluded in a future perspective section.
Author Response
Dear Reviewer,
We appreciate your suggestions very much, we tried to respond to all of them including some new content in the paper, as we consider the suggestions very pertinent to improve the quality of our paper. The changes in the revised manuscript are highlighted in grey.
> This is a paper summarizing existing scientific evidence about the bereavement support interventions for family carers of people who died because of the COVID-19 infection. The review is of good quality, and it is of interest for the readers; however, several minor changes are proposed to improve its quality.
a) In the abstract section, it is not clear how many studies are reporting individual vs. group interventions for such families affected by death of a family member by COVID-19.
b) The introduction section is brief. I would recommend to expand it by adding some research about the efficacy of bereavement groups or individual interventions in other contexts. For instance, there is a high amount of evidence of evidence in the perinatal period and other losts.
RESPONSE:
Thank you for your support and insightful suggestions. The introduction section has expanded with more information about the use of bereavement interventions in other groups or contexts. All interventions are individualized, that information was added in the abstract.
c) In the methods section, the authors reported thay they conducted a scoping review according to Arksey and O'Malley and they describe the five steps and the main research questions. PRISMA guidelines have been updated. I would recommend to use guidelines from 2020. The flow chart diagram is also modified according to the 2020 version. It would improve the quality of the paper.
d) Although there is few data on the topic, the present review is really relevant. Future studies should be designed. The authors, at the end of the discussion section reported that "With these tips in mind, several areas should be studied further". How would the authors design these interventions? Should be implemented in mental health centers, in primary care or in hospitals?
There are few data on the topic. I would recommend to propose an international data source to include all the information about efficacy of these interventions. This can be also concluded in a future perspective section.
RESPONSE
Thank you for your insightful suggestions. We updated the PRISMA flow chart according to new guidelines. We also add more detailed information “Implications and suggestions for research and practice” subsection.
We are deeply grateful for the time and effort you have invested in reviewing our work, and we certainly remain open to any further suggestions for the continued improvement of this paper.
Respectfully,
The Authors
Round 2
Reviewer 1 Report
Reviewer´s second response:
The manuscript has been improved but there is still some editing that needs to be completed.
> Research questions and findings:
Lines 88-101
Regarding the following research questions which are posed:
There needs to be a clearer explanation and more discussion about the findings in relation to this research question in the context of reviewed articles:
"What can we learn from the COVID-19 pandemic about the subsequent impact of this type of death on grief?"
In the manuscript, the word “of” is missing from the above question.
In relation to this research question, this conclusion is stated on lines 350-355.
“While experiencing bereavement is common, the specific experience of loss and bereavement due to the COVID-19 pandemic may have been potentiated by the inability of being near the dying person, bereaving with family and friends, or attending farewell rituals and funeral ceremonies.”
What was found within the reviewed literature in relation to this conclusion?
“What types of interventions to reduce grief and complicated grief during COVID-19 pandemics were addressed in the existing literature?”
The types of interventions are discussed and mentioned in various places within the manuscript: The types of interventions are discussed and mentioned in various places within the manuscript:
Abstract
Line 18: “…bereavement support interventions…”
Lines 25-26: “…Cognitive Behavioural Therapy strategies and other tools to educate, guide, support and promote healthy integration of loss.”
RESPONSE:
Thank you for pointing this out. We agreed with your comment, for that reason some of the learnings that the pandemic brought about the theme of mourning were clarified in the discussion. Additionally, and in the conclusion, the text was reformulated in order to reflect the main results obtained. As for bereavement interventions, these were grouped in the form of psychosocial and psychotherapeutic interventions in order to respond to the diversity of existing strategies. Semantic and grammatical inaccuracies have been corrected.
Reviewer´s response: This has been improved.
---
2) “Although professionals vary in their individual methods and approach, they share common characteristics: non-judgmental compassionate care, as well as a willingness to assist the bereaved people in restoring normal functioning and a perception of socioemotional well-being [11].”
It would be helpful to make reference to the way in which this is stated in the reviewed articles in order to explain how you came to this conclusion.
I also suggest that you coordinate and make clearer within the abstract and introduction which types of interventions are being reviewed.
Reviewer´s response: I see that you have added this sentence. In order to be consistent with the abstract, I would add “psychosocial and psychotherapeutic“:
“Individual and group psychosocial and psychotherapeutic grief interventions have been widely reported in for different populations, including violent loss due to homicide or war and more general grief reactions, such as persistent grief distress following the loss of a long-term spouse, grief associated with profession (e.g. first responders or hospice workers), and unexpected loss (e.g. missing person or perinatal loss) [11] (p.5).”
I think you should put “for” instead of “in”.
Also, leave out the page number because it is not a direct quote.
“In what settings were these interventions provided?”
Reviewer´s response: “All studies analyzed included individual interventions, commonly developed in clinical settings.”
Why do you need to have “individual” in the sentence?
In the research question, you say: “single or grouped interventions, individual or group application”. Keep consistent.
There is a bit of confusion about this generally. I wonder if you mean interdisciplinary approaches for “group applications”. If so, then I would use “interdisciplinary“ rather than “group“ because it can get confused with individual and group therapy. You could also use “each profession” and “integrated professions”.
Information about the settings can be found in Table 2. It would be helpful to provide an overview and summary of the findings for this research question in the conclusion.
“Were there specific components of intervention design (frequency, single or grouped interventions, individual or group application, dose, duration, who delivered intervention, application of the intervention) that subsequently influenced outcomes during bereavement?”
“Were these interventions effective?”
This is discussed in section 3.2.3. where the findings of different studies are reviewed. If the former research question remains as it is, those findings need to be discussed in relation to the factors stated: “frequency, single or grouped interventions, individual or group application, dose, duration, who delivered intervention, application of the intervention.”
RESPONSE:
Thank you for your insightful suggestions. We agree with the reviewer, so we clarify in the abstract which types of interventions are being reviewed and what settings were these interventions provided.
We chose to present a broad discussion on the characteristics of the interventions, given the scarcity of information. Justifying for this reason there is not an individualized answer to each of the questions.
Reviewer´s response: If that is so and you do want to keep all research questions, this needs to be stated in 4.2 within the discussion in the same way as you do here, that this is a weakness to the study or that you were unable to answer all of the research questions because of a scarcity of information. You may want to balance that with some strengths, for instance, that the study concerns the present situation and that there is still a lack of perspective, research and information about how to deal with the aftermath of Covid-19 but the study is a contribution to that field.
----
3) There are a few places within the manuscript where the findings are generalized and not necessarily related to the findings of the review or the connection is not clear. It is suggested to make those general statements more connected to the research questions, literature review and specific findings in that regard.
Lines 246-247
“However, several refer that supporting protective factors for the bereaved’s overall mental health and socioemotional wellbeing are encouraging intervention approaches.”
Which ones are those approaches and how did the authors come to this conclusion?
RESPONSE:
Thank you again for pointing this out. We agreed with your suggestion, and additional information has been added to support this conclusion.
Reviewer´s response: I see that you have added a paragraph (lines 295-300), which is an improvement.
---
4) Lines 260-262
“Interventions should focus on a public health strategy rather than a clinical model, since at-risk persons are not always recognized in advance and people appear reluctant to ask for help.”
In which reference is this stated and how did the author(s) come to this conclusion? The line of thought and logic of the sentence is also difficult to comprehend. “Interventions should focus on a public health strategy rather than a clinical model...” Does this need to be a choice, can’t they go together? I don´t understand the link here that says that it is more important to focus on a public health strategy rather than a clinical model because “... persons are not always recognized in advance and people appear reluctant to ask for help”.
“Interventions should focus on a holistic and integrative model that includes both a public health strategy and a clinical care, since at-risk persons are not always recognized in advance and people appear reluctant to ask for help.”
RESPONSE
Ok! We agreed with your comment, so we reorganized the sentence.
Reviewer´s response: This has not been resolved yet.
In which reference is this stated and how did the author(s) come to this conclusion?
The line of thought and logic of the sentence is still difficult to comprehend. “Interventions should focus on a holistic and integrative model that includes both a public health strategy and a clinical model...”
I don’t understand the logic of the linking here that says that it is more important to focus on a public health strategy rather than a clinical model because “... persons are not always recognized in advance and people appear reluctant to ask for help”.
Kindly explain how this line of logic works so that either I may understand it better and/or you find a clearer way to explain it.
-----
5) Line 296
“Bereaved individuals were approached, mainly, by web-based interventions. This excludes individuals who don’t have access to the internet, and certain groups (e.g. the elderly) weren’t approached by researchers or were reluctant, as the intervention was provided online. This questions whether results are generalizable.”
There are also other reasons why the results may not be generalizable, for example, lack of statistical analysis and few articles being reviewed. I suggest that if the results are not considering generalizability, there is more of a need to go deeper into the function of each method and how it was received by the user.
Reviewer´s response: This has been improved.
----
I question why the three articles which do not yet have findings are included. How do they answer the research questions and contribute to the findings of the manuscript? Why is it important to include those articles? How do they contribute to the findings of the review?
RESPONSE
Thank you for pointing this out. We expanded the limitation subsection with more information that supports the decision of integrating the study protocols. In addition, we added more potential biases to our scoping review.
Reviewer´s response: Yes, this has been improved.
Author Response
Dear Reviewer,
We appreciate your suggestions very much, we tried to respond to all of them including some new content in the paper, as we consider the suggestions very pertinent to improve the quality of our paper. The changes in the revised manuscript are highlighted in grey.
> I see that you have added this sentence. In order to be consistent with the abstract, I would add “psychosocial and psychotherapeutic“:
“Individual and group psychosocial and psychotherapeutic grief interventions have been widely reported in for different populations, including violent loss due to homicide or war and more general grief reactions, such as persistent grief distress following the loss of a long-term spouse, grief associated with profession (e.g. first responders or hospice workers), and unexpected loss (e.g. missing person or perinatal loss) [11] (p.5).”
I think you should put “for” instead of “in”.
Also, leave out the page number because it is not a direct quote.
RESPONSE: Thank you for pointing this out. We agreed with your comment, and we have changed the paragraph.
> “All studies analyzed included individual interventions, commonly developed in clinical settings.”
Why do you need to have “individual” in the sentence?
In the research question, you say: “single or grouped interventions, individual or group application”. Keep consistent.
There is a bit of confusion about this generally. I wonder if you mean interdisciplinary approaches for “group applications”. If so, then I would use “interdisciplinary“ rather than “group“ because it can get confused with individual and group therapy. You could also use “each profession” and “integrated professions”.
RESPONSE: Ok! For a better understanding, we opted to introduce "All studies analysed included interdisciplinary interventions, commonly developed in clinical settings." We maintained consistency with the research question by reformulating it to "Were there specific components of intervention design (frequency, single or grouped interventions, individual or interdisciplinary application, dose, duration, who delivered intervention, application of the intervention) that subsequently influenced outcomes during bereavement?".
> If that is so and you do want to keep all research questions, this needs to be stated in 4.2 within the discussion in the same way as you do here, that this is a weakness to the study or that you were unable to answer all of the research questions because of a scarcity of information. You may want to balance that with some strengths, for instance, that the study concerns the present situation and that there is still a lack of perspective, research and information about how to deal with the aftermath of Covid-19 but the study is a contribution to that field.
RESPONSE: Thank you for your insightful suggestions. It was added as a limitation the inability to answer all investigation questions, however, we balanced that with some strengths.
> This has not been resolved yet.
In which reference is this stated and how did the author(s) come to this conclusion?
The line of thought and logic of the sentence is still difficult to comprehend. “Interventions should focus on a holistic and integrative model that includes both a public health strategy and a clinical model...”
I don’t understand the logic of the linking here that says that it is more important to focus on a public health strategy rather than a clinical model because “... persons are not always recognized in advance and people appear reluctant to ask for help”.
Kindly explain how this line of logic works so that either I may understand it better and/or you find a clearer way to explain it.
RESPONSE: Thanks! We clarify our intention with this statement "Interventions should focus on an integrated care model to capture the components and processes involved. The design and implementation of such interventions need to address people who, in spite of suffering, might feel reluctant in asking for help."
We hope we have accomplished all the suggestions to consider this paper to be published. Thank you again for helping us to improve our work.